A new species of deep-sea grunt, Rhonciscus pauco (Lutjaniformes: Haemulidae), from Puerto Rico

http://orcid.org/0000-0003-4517-9238 Tavera Jose 1 jose.tavera@correounivalle.edu.co
http://orcid.org/0000-0002-0068-4049 Schärer-Umpierre Michelle T. 2
Acero P. Arturo 3
1 Departamento de Biología, Universidad del Valle , Cali , Colombia
2 HJR Reefscaping , Boquerón , Puerto Rico
3 Instituto para el Estudio de las Ciencias del Mar, CECIMAR, Universidad Nacional de Colombia sede Caribe , Santa Marta , Colombia
Robertson D Ross
Electronic publication date: 2022 Jun 2
Publication date: 2022
Volume: 10
Electronic Location ID: e13502
Received 2022 Mar 8; Accepted 2022 May 5
Copyright: © 2022 Tavera et al.
Copyright year: 2022
Copyright holder: Tavera et al.
License: This is an open access article distributed under the terms of the Creative Commons Attribution License, which permits unrestricted use, distribution, reproduction and adaptation in any medium and for any purpose provided that it is properly attributed. For attribution, the original author(s), title, publication source (PeerJ) and either DOI or URL of the article must be cited.
License URL: https://creativecommons.org/licenses/by/4.0/

Keywords: Biodiversity, Inner slope, Ichthyology, West indies, Greater Antilles

Funding: The authors received no funding for this work.

==============================
A fourth species of the genus Rhonciscus (Lutjaniformes: Haemulidae) is described from various specimens collected by small-scale fishers from the insular upper slope of western Puerto Rico. The new species was molecularly recovered as sister to the Eastern Pacific R. branickii, to which it bears many morphological similarities. It is distinguished from other Rhonciscus species by the number of scale rows between the dorsal fin and the lateral line (7), larger and thus fewer scales along the lateral line (48–50), large eyes (9.4–10.4 times in SL), longer caudal peduncle (15.2–20% of SL), larger sized penultimate (14.7–19.1% in SL) and last (7.4–9.5% in SL) dorsal fin spines which translates to a less deeply notched dorsal fin, and its opalescent silver with golden specks live coloration. This grunt, only now recognized by ichthyologists, but well known by local fishers that target snappers and groupers between 200 and 500 m in depth, occurs in far deeper waters than any western Atlantic grunt.

Introduction

The family Haemulidae (together with the snappers, Lutjanidae) is one of the two clades grouped in the order Lutjaniformes Bleeker (Betancur et al., 2017), a tropical lineage that includes commercially important shore fishes. The number of recognized grunt species in two recent checklists varies between 134 and 136, grouped into two subfamilies (Nelson, Grande & Wilson, 2016; Fricke, Eschmeyer & Fong, 2021). Approximately 62 species of the subfamily Haemulinae inhabit New World waters, with the subfamily Plectorhinchinae restricted to African shores and to the Indian and western Pacific Oceans. The most recent revision by Tavera, Acero & Wainwright (2018) recognized 15 New World haemulid genera.

The genus Pomadasys Lacepède, 1802 (type species Sciaena argentea Forsskål, 1775) included several loosely related species from tropical and temperate seas. Species in this polyphyletic assemblage exhibit color and morphological convergence which has resulted in several of them being uncritically assigned to Pomadasys (Tavera et al., 2012). This genus was split into at least five lineages widely spread throughout the family phylogeny, one of which became the genus Rhonciscus (Tavera, Acero & Wainwright, 2018). Further revision is needed to clarify the systematics and taxonomy of the Pomadasys (sensu lato) polyphyletic assemblage.

As for the New World species, Jordan & Evermann (1896) described two genera, Rhencus (type species Pristipoma panamense Steindachner, 1876) and Rhonciscus (type species Pristipoma crocro Cuvier, 1830). These two genera were considered junior synonyms of Pomadasys until Tavera, Acero & Wainwright (2018) resurrected them. The genus Rhonciscus comprises rather elongate species found in marine and brackish waters, but also in rivers and freshwater streams. It presently includes three species, R. crocro, distributed from southern Florida (USA) to at least Rio de Janeiro (Brazil), in the western Atlantic (WA), and two Eastern Pacific (EP) species: R. branickii (Steindachner, 1879) from southern Baja California (Mexico) to Paita (Perú), and R. bayanus (Jordan & Evermann, 1898) from Mazatlán (Mexico) to Rio Tumbes, Perú. Both R. crocro and R. bayanus can be found in freshwater rivers or streams flowing into the ocean. A fourth, unrecognized Rhonciscus species is described based on 14 specimens captured by fishers off the west coast of Puerto Rico, WA.

Materials and Methods

Data collection and analysis

In May 2017 two specimens were collected by the same fisherman from the insular upper slope (280–360 m) off the west coast of Puerto Rico (Tres Cerros, Rincón; 18°20′39″N; 67°17′00″W). Both grunts were caught incidentally on hooks at the end of buoyed vertical line gear, locally known as cala con boya, used by the deep-water, small-scale artisanal fleet. The specimens were deposited at the ichthyology collections of the Sam Noble Museum of Natural History of the University of Oklahoma (Norman, OK, USA) and the Florida Museum of Natural History of the University of Florida (Gainesville, FL, USA). Institutional abbreviations follow Sabaj (2019). Twelve additional specimens were caught in August 2019 by another fisher with Antillean fish traps deployed between 200 and 250 m in depth, approximately 5 km west of the mouth of the Rio Grande de Añasco (Fig. 1). Counts and measurements follow Bussing (1993) and Rocha & Rosa (1999). Measurements were made with calipers and recorded to the nearest 0.1 mm and are presented in Table 1. Information on other Rhonciscus species is included in Table 2 for comparison.

Figure 1 Map of Puerto Rico indicating collection sites of Rhonciscus pauco sp.nov.

Table 1 Meristic and morphometric characters of the holotype, paratype, and 12 additional specimens of the new species Rhonciscus pauco.

Meristics	Holotype	Paratype	1	2	3	4	5	6	7	8	9	10	11	12	
Dorsal fin	XIII, 12	XIII, 12	XIII, 12	XIII, 12	XIII, 12	XIII, 12	XIII, 12	XIII, 12	XIII, 12	XIII, 12	XIII, 12	XIII, 12	XIII, 12	XIII, 12	
Anal fin	III, 7	III, 7	III, 7	III, 7	III, 7	III, 7	III, 7	III, 7	III, 7	III, 7	III, 7	III, 7	III, 7	III, 7	
Pectoral fin rays	16	17	15	16	15	15	16	16	16	16	16	15	15	16	
First arch gill-rakers (upper + lower limb)	5 + 11	5 + 11	5 + 11	5 + 11	5 + 11	5 + 11	5 + 11	5 + 11	5 + 11	5 + 11	5 + 11	5 + 11	5 + 11	5 + 11	
Lateral line scales	50	50	50	50	50	50	50	50	49	48	48	50	50	50	
Morphometrics (mm)															
Total length	266	305	280	283	237	234	240	275	240	233	244	250	243	235	
Standard length	229	244	240	250	195	198	205	237	205	199	210	215	206	199	
Head length	78.8	90.9	72	82.7	60.8	63.7	63.3	77.7	72.5	68.3	68	69.9	69.3	61.8	
Eye diameter	23.5	24.8	23	24.3	19.6	20	20.5	22.9	20.6	20.1	19.8	21	21.4	19.2	
Snout length	22.3	28	21	20.3	15.1	15.6	18.3	22	18.6	19.5	18.8	17.2	17.9	15.1	
Maxillae length	21	24.8	24	23.6	20.2	18.3	19.8	22.3	20	21.4	18	19.8	21.4	17.4	
Interorbital length	18.8	22.1	34	28.5	21.3	21.8	21	26.7	19.5	16.3	20	19.5	20.8	16.9	
Maximum depth	73.9	91.2	80.8	83.7	70.5	68.3	67.4	80.3	69.1	63.7	68.1	68.6	67.5	67.3	
Caudal peduncle depth	22.7	23.4	24.5	24.2	20.4	20.7	20.8	24.7	21.4	20.5	19.6	24	22.8	22.5	
Caudal peduncle length	40.8	42.9	39.8	47.4	36.3	33.3	39.4	47.4	39.1	36	32	36	40.4	30.3	
Predorsal length	72.6	85	74.4	83.4	60.4	60.8	65.8	79.2	68.8	70.9	67.1	73.4	67.3	63.4	
Preanal length	156	174	172	172	139	143	143	163	144	142	147	154	148	140	
Prepelvic length	92.9	95	87.6	94.6	75	78.4	80.3	90.2	81.2	76.4	77.7	81.5	76.6	73.7	
Prepectoral length	78	85.8	76.8	85.9	62	70.2	68.3	79.4	78.8	67	73.1	71.5	71.5	67.3	
Pectoral length	65.5	79.4	74.2	75.6	59.5	61.8	57.2	68.4	59.3	59.6	61.7	60.6	59.6	56.7	
First anal spine	24.1	16.8	15.7	12.7	17	14.3	15.3	15	13.9	14.9	12.1	15.3	12.7	12.4	
Second anal spine	37.6	47.2	44.2	45.8	38.5	40.7	41.2	45.7	40.3	39.8	42.4	41.9	41	43.4	
Third anal spine	25.1	29.8	27.9	25.9	27.1	24.9	25.7	29.4	26.7	23.7	22.4	28.9	24.3	27.7	
Pelvic length	47	55.3	48.5	45.7	39.3	43.1	37.3	46.1	42	37.4	41.9	43.3	39.3	42.2	
Pelvic length including filament	53.1	61.1	53.9	53.9	43	45.7	46	52.7	46.7	46.8	48.1	45	47.1	46	
First dorsal spine	10.9	11.8	11.4	7.3	11.3	9.7	10.2	9.4	10.3	8.3	10.3	8.1	8.2	9.2	
Fourth dorsal spine	37.5	42.3	36.8	38.5	37.2	35.8	36.1	28.6	37	34.4	34.6	33.6	32.4	35	
Fifth dorsal spine	38.9	43.7	36.3	37.7	37.3	35.9	33.1	36.8	35.3	33.5	35.5	31.6	32.6	35.8	
Penultimate dorsal spine	18.2	22.4	20.3	19.9	18	17.4	18.7	17.5	18.2	15.9	18.8	18.5	16.7	19	
Last dorsal spine	18.1	22.9	17.2	19.7	17.8	17.2	18.3	18.1	18	17.1	19.4	19.1	16.2	16.4	
First dorsal ray	25.9	31.7	25.9	30.8	24.5	26.6	26.1	31.8	27.4	25.7	25.7	28.2	24.7	22.7	
Anterior narine	6	5.7	4	5.8	5.1	4.4	4.6	5.5	4.8	3.9	3.9	5.4	4.4	4.2	
Posterior narine	3.6	3.7	3	4.4	2.5	2.5	3.3	3.2	2.9	2.9	2.5	2.8	2.1	2.5	

Table 2 Meristic and morphometric character comparison of all valid Rhonciscus species.

	R. pauco	R. crocro	R. bayanus	R. branickii	
	n = 14	n = 5	n = 4	n = 5	
Ocean	WA	WA	EP	EP	
Meristics					
Dorsal fin	XIII, 12	XIII, 11–12	XII+I, 12	XIII, 12–13	
Anal fin	III, 7	III,7	III, 7–8	III-7	
Pectoral fin	15–16 (17)1	16–18	16–17	16–17	
Lateral line scales	48–50	66–72	63–65	53–55	
Rows of scales above LL	7	8	8	6	
Range of SL (mm)	195–250	176–332	147–205	159–234	
% of SL	Range	Range	Range	Range	
Head length	30–37.25	29.41–37.52	30.41–31.53	28.49–35.57	
Eye diameter	9.42–10.38	6.69–7.81	7.17–8.1	7.09–8.97	
Snout length	7.58–11.47	7.35–10.88	7.7–8.84	5.52–7.81	
Maxillae length	8.57–10.75	8.4–11.23	10.57–12.36	6.11–9.21	
Maximum depth	31.9–37.37	28.79–32.95	27.54–33.8	31.97–36.28	
Caudal peduncle depth	9.33–11.3	9.57–11.35	9.71–10.3	8.68–10.57	
Caudal peduncle length	15.22–20	11.09–14.85	11.88–16.33	14.03–15.5	
Predorsal length	30.7–35.62	30.93–39.35	32.23–33.9	31.19–34.63	
Preanal length	68.12–72.22	71.93–76.23	67.4–71.93	70.53–74.19	
Prepelvic length	36.5–40.56	35.59–43.48	35.64–36.54	34.4–39.84	
Prepectoral length	31.79–38.43	30.11–38.22	29.72–31.93	29.88–34.35	
Pectoral length	27.9–32.54	22.07–24.15	20.21–23.69	28.89–33.66	
First anal spine	5.08–10.52	5.35–8.16	5–8.74	3.29–6.3	
Second anal spine	16.41–21.8	10.82–20.34	16.66–20.01	15.4–17.21	
Third anal spine	10.36–13.91	8.78–11.84	9.12–15.92	9.87–11.56	
Pelvic length	18.19–22.66	18.28–22.36	16.49–22.24	21.56–25.74	
First dorsal spine	2.92–5.79	2.53–3.81	2.46–3.87	2.84–4.81	
Fourth dorsal spine	12.06–19.07	10.55–16.04	12.05–14.12	13.65–17.32	
Fifth dorsal spine	14.69–19.12	11.56–14.88	11.26–14.88	13.02–16.24	
Penultimate dorsal spine	7.38–9.54	3.63–6.03	5.08–5.66	4.06–6.66	
Last dorsal spine	7.16–9.38	4.51–7.38	5.61–7	4.85–6.73	
First dorsal ray	10.79–13.43	8.8–11.44	7.46–11.73	8.11–12.11	
Anterior narine	1.66–2.62	0.69–0.78	0.32–0.64	0.79–1.01	
Posterior narine	1.01–1.76	0.36–1.45	0.53–0.94	1.09–1.61	
Times in SL					
Head length	2.68–3.33	2.67–3.4	3.17–3.29	2.81–3.51	
Eye diameter	9.62–10.6	12.8–14.94	12.35–13.94	11.15–14.11	
Snout length	8.71–13.17	9.19–13.61	11.31–12.98	12.81–18.13	
Maxillae length	9.29–11.66	8.91–11.9	8.09–9.46	10.86–16.38	
Maximum depth	2.67–3.13	3.04–3.47	2.96–3.63	2.76–3.13	
Caudal peduncle depth	8.84–10.71	8.81–10.45	9.71–10.3	9.46–11.53	
Caudal peduncle length	5–6.56	6.73–9.02	6.12–8.42	6.45–7.13	
Predorsal length	2.8–3.25	2.54–3.23	2.95–3.1	2.89–3.21	
Preanal length	1.38–1.46	1.31–1.39	1.39–1.48	1.35–1.42	
Prepelvic length	2.46–2.73	2.3–2.81	2.74–2.81	2.51–2.91	
Prepectoral length	2.6–3.14	2.62–3.32	3.13–3.36	2.91–3.35	
Pectoral length	3.07–3.58	4.14–4.53	4.22–4.95	2.97–3.46	
First anal spine	9.5–19.68	12.25–18.7	11.44–20.01	15.88–30.42	
Second anal spine	4.58–6.09	4.92–9.24	5–6	5.81–6.49	
Third anal spine	7.18–9.65	8.45–11.39	6.28–10.97	8.65–10.13	
Pelvic length	4.41–5.49	4.47–5.47	4.5–6.07	3.88–4.64	
First dorsal spine	17.25–34.24	26.24–39.46	25.85–40.69	20.79–35.25	
Fourth dorsal spine	5.24–8.28	6.23–9.48	7.08–8.3	5.77–7.33	
Fifth dorsal spine	5.22–6.8	6.72–8.65	6.72–8.88	6.16–7.68	
Penultimate dorsal spine	10.47–13.54	16.58–27.55	17.66–19.69	15.01–24.63	
Last dorsal spine	10.65–13.95	13.56–22.18	14.28–17.82	14.87–20.61	
First dorsal ray	7.44–9.26	8.74–11.36	8.52–13.4	8.26–12.33	
Anterior narine	38.16–60	128.07–145.12	155.83–311.63	99.21–126.91	
Posterior narine	56.81–98.09	68.78–277.59	105.94–189.54	62.13–92	
Times in HL					
Eye diameter	3.08–3.66	3.9–5.17	3.89–4.28	3.65–4.02	
Snout length	3.24–4.09	3.45–4.21	3.44–4.09	4.41–5.27	
Maxilla length	3–3.77	2.92–3.57	2.46–2.94	3.86–4.69	
Pectoral length	1.63–2.21	1.98–3.25	2.04–2.77	1.76–2.73	
Fifth dorsal spine	3.26–6.51	3.73–7.02	3.48–6.31	4.55–8.67	
Anterior narine	11.92–18	37.66–51.62	47.38–95.69	28.27–37.56	
Posterior narine	18.79–33	25.81–81.63	32.21–58.2	17.7–32.73	

Tissue samples were taken from the pectoral fin of the holotype and paratype specimens and stored in 96% EtOH. Molecular information is provided as additional evidence of the evolutionary distinction of this new taxon. DNA extraction and COI PCR amplification details can be seen in Tavera et al. (2012). Sequencing was performed in one direction on an ABI 3100 automated sequencer (Applied Biosystems, Foster City, CA, USA). The sequences available in GenBank for all species of the genus Rhonciscus plus Haemulon sciurus (outgroup) were used for molecular comparisons and a preliminary phylogenetic reconstruction. GenBank accession numbers are as follows: R. bayanusMF446583; R. branickiiJQ741307, JQ741308, MF956957, HQ676794; R. crocroJQ741309; and H. sciurusEU697541. COI sequences were cleaned and trimmed with Geneious 9.1.8 (Kearse et al., 2012) and Muscle (Edgar, 2004) was used as the alignment algorithm with default parameters. Phylogenetic relationships were assessed using the Maximum Likelihood (ML) method. Five independent runs with a random starting tree, GTRGamma model of nucleotide substitution, 1,000 bootstrap analysis and search for the best scoring ML tree were performed with RAxML GUI 0.93 (Silvestro & Michalak, 2012; Stamatakis, 2014). Corrected intraspecific and interspecific genetic distances were calculated for all available sequences.

Ethical approval

“All applicable international, national, and/or institutional guidelines for the care and use of animals were strictly followed. All animal sample collection protocols complied with the current laws of Puerto Rico.” Specimens were collected under the authorization of the Puerto Rico Department of Natural and Environmental Resources research permit # 2017-IC-031 granted to M. Schärer-Umpierre. “The electronic version of this article in Portable Document Format (PDF) will represent a published work according to the International Commission on Zoological Nomenclature (ICZN), and hence the new names contained in the electronic version are effectively published under that Code from the electronic edition alone. This published work and the nomenclatural acts it contains have been registered in ZooBank, the online registration system for the ICZN. The ZooBank LSIDs (Life Science Identifiers) can be resolved and the associated information viewed through any standard web browser by appending the LSID to the prefix http://zoobank.org/. The LSID for this publication is: (urn:lsid:zoobank.org:act:6BF57CB4-60DE-4FCE-BF33-B02DF8E7FE05). The online version of this work is archived and available from the following digital repositories: PeerJ, PubMed Central SCIE and CLOCKSS.”

Results

Rhonciscus pauco sp. nov.

urn:lsid:zoobank.org:act:6BF57CB4-60DE-4FCE-BF33-B02DF8E7FE05

Opalescent Grunt

(Spanish name: Ronco opalescente)

Figures 2 to 5

Figure 2 Rhonciscus pauco, sp. nov. Sam Noble Oklahoma Museum of Natural History OMNH 86864, holotype, 266 mm SL, from Tres Cerros, Rincón, Puerto Rico.

Holotype. OMNH 86864, 266 mm SL. Tres Cerros, Rincón, Puerto Rico, 18.34433°N; 67.28343°W. Collected with hook and line at 280 m depth. May 1, 2017.

Paratype. UF 242681, 305 mm SL. Tres Cerros, Rincón, Puerto Rico, 18.34433°N; 67.28343°W. Collected with hook and line at 360 m depth. May 5, 2017.

Diagnosis. A species of the genus Rhonciscus with XIII, 12 (total 25) dorsal-fin rays; anal-fin rays III, 7; pectoral-fin rays 15–16, 17(1); rather elongate body, maximum depth 32–37.4% SL; convex predorsal profile; eye large, its diameter 9.4% to 10.4% SL; snout subequal to eye, its length 7.6% to 11.5% SL; very coarse serrations on angle of preopercular margin; pectoral fin long (28–32.5% SL) extending beyond the tip of pelvic fin, barely reaching anus; head length 30–37.3% SL; longest dorsal-fin spine (fifth) (12.1–19.1% SL); relatively long and much thicker second anal-fin spine (16.4–21.8% SL), long caudal peduncle (15.2–20% of SL), and a large size of the penultimate (14.7–19.1% in SL) and last dorsal-fin (7.4–9.5% in SL) spines which translate to a less deeply notched dorsal fin, eye diameter 0.5 to 0.6 times length of anal fin spine; maxilla reaching anterior border of pupil; seven scale rows between dorsal fin and lateral line; 48 to 50 lateral–line scales.

Description. Measurements are presented as percentage of SL in Table 2. Body elongate, with greatest depth at vertical through origin of pelvic fins (2.7 to 3.1 times in SL); predorsal profile is slightly convex with a tenuous depression above the eye; head length 2.7 to 3.3 in SL; eye diameter 9.6–10.6 times in SL and 3.1–3.7 times in head length (HL); interorbital distance 2.1–4.2 in HL; snout slightly convex, length 3.3–4.1 in HL; mouth terminal and moderately oblique; maxilla reaches to or slightly beyond anterior margin of pupil, 3 to 3.8 in HL: maxilla covered by suborbital when mouth is closed; upper and lower jaws with small conical teeth; conspicuously serrated preopercle (Fig. 4A); mouth strongly protrusible; one to two rear chin pores, well-separated (Fig. 4B); total number of first arch gill-rakers 16, upper limb 5, lower limb 11; anterior narine 12 to 18 in HL: posterior narine 18.8 to 33 in HL; pectoral-fin length 1.6 to 2.2 in HL; predorsal-fin length 2.8 to 3.3 in SL; prepectoral-fin length 2.6 to 3.2 in SL; prepelvic-fin length 2.5 to 2.8 in SL; preanal-fin length 1.4 to 1.5 in SL; fifth dorsal-fin spine 3.3 to 6.5 in HL; second anal-fin spine 4.6 to 6.1 in SL; depth of caudal peduncle 8.8–10.8 times in SL; length of caudal peduncle 5–6.6 in SL; length of penultimate dorsal spine 5.3–6.8 times in SL and last dorsal spine 10.5–13.5 times in SL; lateral-line scales 48–50; scale rows between lateral line and dorsal fin 7; no scales present on anal or dorsal fin membranes (Fig. 4C); pelvic fin with elongate filament on first ray (Fig. 4D).

Color in life. Body opalescent silver with golden specks; some of the spines reflect iridescent blue in direct light. Opaque to translucent dorsal, anal, fins and dusky caudal fin (Fig. 3). Two small dark spots present on the trailing edge of the lower lobe of the caudal fin. A black marking across the marginal edge of the spine dorsal fin membrane (Fig. 4C). Brown to blackish spots over the attachment of some scales, especially towards the back of the trunk (Fig. 4C). Figure 5 is an underwater photograph taken with a ROV at 218 m depth in western Puerto Rico where two specimens of the new species, including their live coloration, can be seen.

Figure 3 Rhonciscus pauco, sp. nov. Fresh coloration.

Figure 4 Rhonciscus pauco, sp. nov.

(A) Detail of the head, especially on preopercle serration. (B) Rear chin pores. (C) Spine dorsal fin membrane and dark spots over scales. (D) Pelvic fin with filament on the first ray.

Figure 5 Rhonciscus pauco, sp. nov. Underwater photograph taken at 218 m depth in western Puerto Rico.

Image credit to NOAA NCCOS 2022.

Distribution. Rhonciscus pauco is found on the deep shelf and upper slope of the western coast of the northeastern Caribbean island of Puerto Rico. We are uncertain of the species’ exact range, but fishers report capturing them exclusively in fine sediment habitats distributed between the municipalities of Rincón and Mayagüez, off western Puerto Rico (Fig. 1). No additional information is currently available.

Habitat. Collection depths range from 200–360 m in fine unconsolidated sediment or mud habitats (Fig. 5).

Etymology. The name pauco comes from the fisher’s nickname Paúco, Edwin Font, who already knew of this fish locally called burro or ronco (grunt). Mr. Font was the first to report and provided specimens to MS, although it is recognized by various fishers as a component of the deep-water catch in western Puerto Rico.

Comparisons. Rhonciscus pauco is morphologically very close to its sister EP species R. branickii with which many measurements and counts overlaps. However, it can be distinguished from the three species of the genus, the WA R. crocro, and both EP species, R. bayanus and R. branickii, by the seven scale rows between the dorsal fin and the lateral line; R. branickii has five according to McKay & Schneider (1995) or six (our data), while R. bayanus and R. crocro have eight scale rows above the lateral line. It differs clearly from its WA congener by having 48–50 lateral–line scales (vs 66–72 in R. crocro) and 11 short lower gill rakers (vs 7–9 in R. crocro). Our scale counts for R. crocro differ from those presented by Lindeman & Toxey (2002), who report five and six rows between the dorsal fin and the lateral line and 53–55 lateral–line scales as opposed to eight rows and 66–72 lateral–line scales. In both cases the new species can be unambiguously distinguished. The new species has a coarse serrated preopercle, with 2–3 large spines at the corner and 20–21 relatively small spines on the vertical border; the preopercle is weakly or not serrated in R. bayanus. Rhonciscus pauco has much larger eyes (9.4–10.4% of SL) compared to 7.2–8.1% in R. bayanus, 7.1–8.9% in R. branickii, and 6.7–7.8% in R. crocro. Bussing (1993) reported that the eye diameter of R. branickii ranged between 2.9 and 3.8 times in head length while McKay & Schneider (1995) published 3.1–3.4; Meek & Hildebrand (1925) reported larger eyes (2.6 to 3.2) based on specimens smaller than 15 cm. Our data for R. pauco (Table 2: 3.08–3.66) overlap such information; however, the lack of size data in Bussing (1993) and McKay & Schneider (1995) precludes a more detailed comparison. With our data, that are size comparable, R. pauco can be clearly separated from R. branickii.

Furthermore, R. pauco has a longer caudal peduncle (15.2–20% of SL) compared to 11.9–16.3% in R. bayanus, 14.0–15.5% in R. branickii and 11.1–14.9% in R. crocro. The larger size of the penultimate (14.7–19.1% in SL) and last dorsal-fin (7.4–9.5% in SL) spines translate to a less deeply notched dorsal fin in R. pauco. Additionally, similarly sized R. pauco can be distinguished from its EP sister species R. branickii by having longer snouts and longer maxilla (3.24–4.09 and 3–3.77 in HL in R. pauco vs 4.41–5.27 and 3.86–4.69 in R. branickii). A detailed comparison of morphometric proportions is included in Table 2. Life coloration is another feature separating the new Puerto Rican species from R. branickii; R. pauco is light silver to golden in color, with opalescence, while its sister species EP is dull dark brown, without bright colors or opalescence, and with a darker marking along the entire opercular margin.

Discussion

The unexpected discovery of a new species of grunt at this depth on the Puerto Rican slope is an issue of biogeographic interest in the Caribbean. For snapper and grouper fishermen in deep waters off the west coast of Puerto Rico, this species is well known, despite having gone unnoticed and not being formally described until now. R. pauco is captured incidentally with the snappers Etelis oculatus and Pristipomoides macropthalmus but discarded due to its low market value. It is commonly referred to as burro, ronco or viejo, the latter name also used by some fishers to identify the congener grunt, R. crocro, that is not rare in shallow, brackish waters of large volume rivers throughout Puerto Rico (Engman et al., 2019).

Other than a new record of a single specimen of R. branickii (USNM 422650) collected in the eastern Pacific off Panama at 430 to 500 m, only six Caribbean grunt species occur at one hundred meters depth or below, with 120 m as the maximum known depth (Robertson et al., 2019). The new species is found inhabiting waters deeper than 200 m where there is extremely low light penetration.

What is even more surprising is that two of the four Rhonciscus species are known to occupy very shallow, often brackish or freshwater environments. The type species of the genus, R. crocro, inhabits waters shallower than 20 m and is frequently found in the lower parts of short, rapid rivers or slow creeks in eastern Florida (KC Lindeman, 2022, personal communication) as well as in 1 m or less in depth in the Rio Grande de Añasco where it flows into the ocean (M Schärer-Umpierre, 2020, personal observation). The two EP Rhonciscus species are also reported in shallow water, especially R. bayanus, known at depths less than 10 m (Robertson & Allen, 2015). Rhonciscus branickii, on the other hand, has been reported once at a depth of 500 m; this species is very similar to R. pauco and is considered its sister species. However, R. branickii is an EP species. Besides this deep record no other grunt of this genus been reported from depths at which R. pauco has been caught. The evolutionary processes that led to the existence of this deep–water grunt in the northeastern Caribbean, far away from the continental shores of Central and South America with its WA congener known primarily from coastal rivers, are of utmost biogeographical interest.

Finally, the phylogenetic tree (Fig. 6), which includes all valid species of the genus, resulted in a topology with R. pauco placed as a sister species to the EP R. branickii, with a sequence divergence of 1.7–2.3%, establishing a new trans-isthmian sister-pair within the family Haemulidae. Remaining COI genetic distances of R. pauco are 13.6–14% with R. bayanus and 12.7–13.2% with R. crocro. The distance between these geminate species across the isthmus of Panama is relatively small (1.7–2.3%) yet consistent with values of two other trans-isthmian pairs in the genus Anisotremus (2.0–2.4%) (Tavera et al., 2012). The COI genetic distances in seven trans-isthmian sister pairs of Haemulidae range from 2.0% between Anisotremus surinamensis (WA) and A. interruptus (EP) to 18% between Genyatremus cavifrons (WA) and G. pacifici (EP). Interspecific distance values near 2% divergence have been reported in both freshwater and marine fishes (Ward et al., 2005; Shen et al., 2016). In any case, interspecific genetic distances between R. pauco and R. branickii are larger than the intraspecific genetic distances (R. pauco: 0% and R. branickii: 0.2–0.8%).

Figure 6 COI maximum likelihood phylogenetic tree of the species Rhonciscus, including the new species Rhonciscus pauco.

Numbers over branches correspond to the bootstrap support.

It is common to find variable intra- and interspecific distances and defining a threshold becomes difficult as the mutation rate varies considerably among species. A small distance may indicate very strong differentiation between sister species that have diverged comparatively recently, while COI will not detect “substantial” differentiation. Perhaps this marker in this group evolves quite slowly despite the irrefutable fact that the new species is reproductively isolated, adapted to different environmental conditions, yet resembling its EP sibling species. The magnitude of variability in the mtDNA divergence between all pairs of trans-isthmic grunts is consistent with an asynchronous divergence, as described for other taxa (Knowlton & Weigt, 1998; Leigh, O’Dea & Vermeij, 2014; O’Dea et al., 2016).

Additional material examined:

Rhonciscus pauco: 240 mm SL, mouth of the Rio Grande de Añasco, Puerto Rico 2021. 250 mm SL, mouth of the Rio Grande de Añasco, Puerto Rico 2021.195 mm SL, Puerto Rico 2021. 198 mm SL, mouth of the Rio Grande de Añasco, Puerto Rico 2021. 205 mm SL, mouth of the Rio Grande de Añasco. Puerto Rico 2021. 237 mm SL, mouth of the Rio Grande de Añasco, Puerto Rico 2021. 205 mm SL, mouth of the Rio Grande de Añasco, Puerto Rico 2021. 199 mm SL, mouth of the Rio Grande de Añasco, Puerto Rico 2021. 210 mm SL, mouth of the Rio Grande de Añasco, Puerto Rico 2021. 215 mm SL, mouth of the Rio Grande de Añasco, Puerto Rico 2021. 206 mm SL, mouth of the Rio Grande de Añasco, Puerto Rico 2021. 199 mm SL, mouth of the Rio Grande de Añasco, Puerto Rico 2021.

Rhonciscus bayanus: CIRUV 186.2 mm SL. Gorgona Island, Colombia, 2018. Photograph 185.6 mm SL. Chiapas, Mexico, 2011. Rhonciscus branickii: CICIMAR 159.2 mm SL. Piura, Peru, 2009. CICIMAR 234.4 mm SL. Baja California, Mexico, 2010. CICIMAR 191.2 mm SL. Baja California, Mexico, 2010. CICIMAR 193.7 mm SL. Baja California, Mexico, 2010. CIRUV 170.4 mm SL. Gorgona Island, Colombia, 2018.

Rhonciscus crocro: Holotype MNHN 733 185.7 SL, Martinique Island, West Indies, 1830. CICIMAR 220.4 mm SL. Buritaca, Colombia, 2010. INVEMAR 245.5 mm SL. Ciénaga Grande, Colombia, 2010. INVEMAR 176.7 mm SL. Buritaca, Colombia, 2010. Photograph 332.1 mm SL. Rio Grande de Añasco, Puerto Rico, 2018.

Conclusions

Based on morphology, geographic distribution, and molecular analyses, this study describes as a new species the first known western Atlantic deep-sea-dwelling grunt collected from the upper Puerto Rican slope. This finding is an issue of evolutionary and biogeographic interest for the family, which is composed of shallow water species, commonly found on upper shelf waters up to 100 m in depth with a strong preference for shallow environments of less than 40 m. Moreover, the fact that some species of the genus Rhonciscus are frequently found in estuaries, rivers and freshwater streams makes this finding quite remarkable.

Supplemental Information

Supplemental Information 1 Raw measurements and meristics.

Click here for additional data file.

Supplemental Information 2 COI partial mitochondrial DNA sequence for Holotype of the new species Rhonciscus pauco.

Click here for additional data file.

Supplemental Information 3 COI partial mitochondrial DNA sequence for Paratype of the new species Rhonciscus pauco.

Click here for additional data file.

We are indebted with E. Font for the initial collection of specimens and for sharing the traditional ecological knowledge of this fish and its habits, as well as H. Vargas that provided additional specimens that led to the scientific recognition of this new species of grunt. The authors are extremely appreciative of A. Veglia and N. Schizas of the Department of Marine Sciences of the University of Puerto Rico for initial advice. K. Overly and T. Battista were instrumental in obtaining the photo from a NOAA NCCOS 2022 ROV dive off western Puerto Rico. Contribution No. 19 of the Instituto de Ciencias del Mar y Limnología de la Universidad del Valle, INCIMAR. Contribution No. 539 of the Instituto de Estudio de las Ciencias del Mar, CECIMAR.

Additional Information and Declarations

Competing Interests

Author Contributions

Field Study Permissions

DNA Deposition

Data Availability

New Species Registration

The authors declare that they have no competing interests. Michelle Schärer is employed by HJR Reefscaping.

Jose Tavera conceived and designed the experiments, analyzed the data, prepared figures and/or tables, authored or reviewed drafts of the article, molecular lab work, and approved the final draft.

Michelle T. Schärer-Umpierre conceived and designed the experiments, prepared figures and/or tables, authored or reviewed drafts of the article, and approved the final draft.

Arturo Acero conceived and designed the experiments, authored or reviewed drafts of the article, and approved the final draft.

The following information was supplied relating to field study approvals (i.e., approving body and any reference numbers):

Puerto Rico Department of Natural and Environmental Resources # 2017-IC-031.

The following information was supplied regarding the deposition of DNA sequences:

The sequences of the new species are available at GenBank: OM904565 and OM904566.

The following information was supplied regarding data availability:

The raw data is available in the Supplemental File.

The following information was supplied regarding the registration of a newly described species:

Publication LSID: urn:lsid:zoobank.org:pub:8DA659A3-D402-4F59-A53F-334E6D39A71A.

Rhonciscus pauco Tavera, Schärer & Acero P., 2022 species LSID: urn:lsid:zoobank.org:act:6BF57CB4-60DE-4FCE-BF33-B02DF8E7FE05.

Rhonciscus Jordan & Evermann, 1896 sec. Tavera, Schärer & Acero P. 2022 genus LSID: urn:lsid:zoobank.org:act:6D77F4E0-C975-408B-98A6-3C7F4EFB466F.

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
