# Peer review of "A new species of deep-sea grunt, Rhonciscus pauco (Lutjaniformes: Haemulidae), from Puerto Rico"

_PeerJ, doi:10.7717/peerj.13502_

## Round 0.1 · original submission · Major Revisions

Reviewer one particularly requires significant changes. Many of those seem quite justified. Reviewer two contributed little in the way of revision requirements
Here are my comments, in no particular order:

1. You refer to differences between your scale counts for R crocro and those given in Robertson et al 2019 (lines 183-185). That information is incomplete. The data in that website are derived from data in the Haemulid chapter of FAO TNWA: Lindeman & Toxey 2002. Where those authors obtained is unclear from their literature cited. You need to review all sources of taxonomic info to check scale counts that.Eschmeyers Catalog of Fishes refers to, which are not hard to find. Your specimens have 8 rows above and 66-72 on the lateral line…the former agrees with Bean 1884, but not the latter who gives 50. Also McEachran gives 50. And Miller 2006 gives other numbers. You need to include and discuss all this variation in R crocro data

2. Scalation of fins is not mentioned in description of pauco; it is in other descriptions of congeners. You should include this.

3. There are no standard or total lengths of any specie given in table 2, which is strange. For all any reader knows you are comparing meritics of fish of different sizes in the different species, and differences could be due to allometric growth.

4. To what extent do the briefly presented diagnostic features of Rhonciscus given in Tavera et al 2018 need revision in the light of differences between pauco and the other species? Dorsal spines and pectoral length of R pauco seem different, as is habitat, snout length and preopercle spination? Eye size? I suggest a revised definition of Rhonciscus incorporating variation among all four congeners.


5. The materials and methods need a data-availability statement

6. The English and grammar both need attention….try using the free grammar-checking program https://www.grammarly.com/ to flag issues. There also are some Spanish terms used that are not English…check the entire ms for this.

Reviewer 1 ·

Basic reporting

NC

Experimental design

fails to meet standards, see additional comments

Validity of the findings

fails to meet standards, see additional comments

Additional comments

This ms describes a new species of deep water grunt in the caribbean, the sister species of the eastern Pacific species R. branickii.

It is a likely new and technically valid species, however the ms compares it to mainly the clearly unrelated Caribbean congener, instead of the relevant comparison to its sister species. Unfortunately, the sample size is so small that the question is moot, I have no idea how pauco is different from branickii.

The emphasis of the ms, that pauco is not crocro or bayanus, is unneeded, they are completely different and separated in a simple two couplet key. The comparison to the sister species is what is relevant.

The next emphasis of the ms is that pauco is an unusual deep water species in Rhonciscus- but "unusual" is barely applicable in a set of only 4. Then branickii, the fourth is actually found very deep, up to 500m in museum metadata, as well as estuarine. So the genus contains two shallow species, one deep and one both.

The definition of a valid species is whether it can be distinguished from its sister species reliably- not a statement that it is genetically isolated- there are hundreds of reef fish species isolated between sides of the Atlantic, or Caribbean and Brazil, or between the central and eastern Pacific, or between the Indian and Pacific Oceans that are considered the same species.

There are a number of transisthmic populations that are considered the same species (taxonomically if not practically) because there is no reliable diagnostic character separating them. In that case, one would not split the species based only on a genetic difference- people spend a lot of effort trying to find a difference that is persuasive and not an artifact that will disappear with adding a few more specimens. For example, Etropus crossotus is still considered the same species on both sides because after measuring and counting hundreds of fish in a huge study (long time ago), a clean persuasive difference cannot be found. There are others, like Hyporthodus mystacinus.

Thus the relevant comparison is what character distinguishes pauco from branickii- and that should not be done with 5 comparison fishes- I understand the limitations of access these days to museums, but there are large numbers of branickii specimens in major museums, such as SIO & USNM, from all depths from 0-500m. Perhaps a collaboration with a museum staff member who could do the relevant measurements on 100 fish would find the true difference.

The DNA comparison between the two sisters shows they are very close, making a morphological difference much harder to prove and that, by itself, means the sample sizes must be large. The DNA barcode database shows the two species are only 1.5% divergent (minimum interspecific distance, not mean.. two lineages that actually share haplotypes will have a mean difference, even a large one). The few sequences of branickii in the database already have a particularly high intraspecific variation, greater than 1.5% (maximum intraspecific; similarly the mean variation is irrelevant, since that can be low, even in the case of intraspecific haplotypes more distant from each other than is the other species' haplotypes- meaning one cannot reject the hypothesis the two groups are the same). In this pauco/branickii case the result technically violates the "barcode gap"- i.e. min interspecific distance needs to be greater than max intraspecific, but that is not a disqualifier, just an indication one needs to be even more persuasive on the phenotypic comparison. In other words, when genetics do not provide much support for the split, one needs to document the morphological difference more completely.

Other than that overall issue, the methodology for comparing sets of fish specimens is unconvincing. We need to be assess whether measurements are 1) representative (covers geographic range, depth range, size range, etc.); 2) reliable (repeatable and comparable); and 3) robust (sample size).

It is obvious that comparing a set of morphometrics between 10 deep fish in the Atlantic and 10 freshwater fish in the Pacific is not useful. Similarly, 10 fish under 100 mm compared to 10 fish over 200mm, or 10 fish from California to 10 fish from Puerto Rico. Given the difficulty of collecting pauco, a small sample size is adequate if the set it is compared to is a reasonable sample of branickii to see if there is anything pauco have that can single it out from the specimens of branickii.
The main problem in grunts is the question of reliability. If there is unexpectedly more variation out there than is suggested by the few specimens in this study, then it means that feature cannot be established as a diagnostic character. For grunts, the variability in reported morphometrics in the literature is so extreme that either the guides and keys are mostly errors or the variation from one population to another is an order of magnitude more than from a small sampling.

For example, in this ms there are values that are in a different world from values in the guides and keys. The ms mentions that in passing on a comparison with crocro that literature guide has completely different values for crocro than the ms- not even overlapping- raising questions about consistency. But crocro is not an issue, it is different in many features. The pertinent comparison with branickii is needed, but the guide to eastern Pacific shorefishes is in a different world- the eye diameter, the main feature said to distinguish the species, of this ms' five branickii specimens is so different, it is hard to understand: 2.6-3.2 in HL in guide-- vs. 3.95 to 4.0 in HL in the ms (Table 2- HL=28.5-35.6 %SL divided by eye diam=7.1-9 %SL). For a grunt, a range of 3.95 to 4.0 is far too narrow a range to represent a species, and the 5 specimens range from 159 mm to 234 mm- that alone should increase the range of eye diameters (far) beyond 3.95 to 4.0 based on allometry (smaller fish have relatively larger eyes).

There are numerous other not-even-overlapping differences between meristic and morphometric values found in the ms and in the various literature, suggesting there is vastly more variability than encountered in the small samples here.

Note that gross morphometrics, such as eye size or body depth are known to be highly variable, so differences found must be approached with skepticism, especially when the populations are so close genetically. Typically differences between sister species are in color, markings, or meristics, at best.

To understand the enormous variation in reported eye sizes in rhonciscus, I researched a few photographs and find these photographs in the attached document- the eye size of pauco and branickii look the same. Four photos of the grunt P. macracanthus show huge variation in eye diameter, all among adults.


There also seem to be errors- these results seem impossible for the 4 species in Table 2: these measurements below are first pauco range, and then branickii range: branickii have a first dorsal spine almost ten times smaller than pauco? Attached in the photographs one from the guide of a branickii with a normal length first dorsal spine.

First dorsal spine


20.93-25.04


2.84-4.81

Fourth dorsal spine


2.92-5.79


13.65-17.32

Fifth dorsal spine


12.06-19.07


13.02-16.24

Penultimate dorsal spine


14.69-19.12


4.06-6.66

Annotated reviews are not available for download in order to protect the identity of reviewers who chose to remain anonymous.

·

Basic reporting

1. BASIC REPORTING
- Clear, unambiguous, professional English language used throughout.
Yes, very well written.
- Intro & background to show context. Literature well referenced & relevant.
Yes, see general comments.
- Structure conforms to PeerJ standards.
Yes.
- Raw data
I may be me missing something but I cannot find the data deposition statement.
- Figures are relevant, high quality, well labelled & described.
Yes.

Experimental design

- Original primary research within journal scope.
Yes.
- Research question well defined, relevant, meaningful. It is stated how the research fills an identified knowledge gap.
Yes to all of these issues.
- Rigorous investigation performed to a high technical & ethical standard.
Yes, see general comments.
- Methods described with sufficient detail & information to replicate.
Yes.

Validity of the findings

3. VALIDITY OF THE FINDINGS
- Impact and novelty not assessed. Meaningful replication encouraged where rationale & benefit to literature is clearly stated.
Yes, in terms of this as a new species description.
- All underlying data have been provided; they are robust, statistically sound, & controlled.
Yes, as i can determine for a new species description.
- Conclusions
Well written, supported by results, linked to original question.

Additional comments

Review comments from the manuscript:
The paper is more than a good new species description, it indeed reflects a very interesting and novel biogeographical issue in a solid fashion.
- Line 24. After the phrase “sister species”, the authors may want to provide the characters that suggest this relationship, not essential.
- 29. If the authors desired, they could end the abstract by saying that the species occurs in far deeper waters than Western Atlantic congeners.
- 44-46. For the uninformed, this paragraph does not refer to Rhonciscus. It refers to Pomadasys being split into lineages – seems like there should be a statement saying that 1 of those lineages was Rhonciscus.
- 76. “Of” should probably be “on”.
- 165-167. This is an extremely limited area. Have fishers elsewhere on the island been asked about this species?
- 168. Is the phrase “ that suggests extending these limits.” really needed? Perhaps just take out the phrase in quotes, it seems a little early to imply this is the only population. I wonder if any are at Mona Island or the eastern Dominican Republic across the Mona Channel.
- Discussion. There are 2 long and dense paragraphs in this section. They are solid but would be easier to read if split into smaller paragraphs, suggestions follow.
- 207. Suggest inserting a new paragraph before “What is even more surprising…”
- 210. After rivers suggest inserting: “… or slow creeks (in easter Florida).”
- 217. After South America possibly insert: “… with it’s WA congener known primarily from coastal rivers, …”
- 229. Suggest new paragraph before “It is common…”
- Tables and captions are solid.
- Figure 4 caption: first line, “on” could be “of” or it could best, be deleted.
- Figure 5 is valuable. Fyi, the top fish image is laterally out of alignment (a bit) with the three images below.

---

## Round 0.2 · accepted · Accept

The paper provides a satisfactory description of a new deepwater grunt, Rhonciscus pauco, that has been discovered off western Puerto Rico. It is a member of a genus with only three other, well known species. Its only other Caribbean area congener is restricted to very shallow water and rivers and has a very different morphology, as quantified in this description. The only other Rhonciscus species found in deep water is R paucus’ sister species in the East Pacific. Those two clearly differ in a number of quantified morphological characteristics and coloration, as well as being genetically distinct. The level of genetic difference between those two Rhonciscus is similar to that among some transisthmian sister pairs in other genera in the same family. There are no significance issues with this description, which is much improved over the first version.

There are a few minor issues with the text that need correcting:
I had to send those as a comments pdf since these comment boxes do not allow formatted text